# Unveiling the Structural Characteristics and Bioactivities of the Polysaccharides Extracted from Endophytic *Penicillium* sp.

**DOI:** 10.3390/molecules28155788

**Published:** 2023-07-31

**Authors:** Kumar Vishven Naveen, Anbazhagan Sathiyaseelan, Sumana Mandal, Kiseok Han, Myeong-Hyeon Wang

**Affiliations:** 1Department of Bio-Health Convergence, Kangwon National University, Chuncheon 24341, Republic of Korea; naveen.vishven@gmail.com (K.V.N.); sathiyaseelan.bio@gmail.com (A.S.); seq0120@gmail.com (K.H.); 2Department of Chemistry, Sungkyunkwan University, Suwon 16419, Republic of Korea; sumanamandal3@gmail.com

**Keywords:** endophytes, *Penicillium*, polysaccharides, oxidative stress, prostate cancer, natural product

## Abstract

Polysaccharides are abundantly present in fungi and are gaining recognition for their exceptional bioactivities. Hence, the present study aimed to extract intracellular polysaccharides (IPS-1 and IPS-2) from the endophytic *Penicillium radiatolobatum* and compare their physicochemical and bioactive attributes. The monosaccharide composition analysis revealed the existence of galactose, glucose, and mannose in both the IPS, while a trace amount of xylose was found in IPS-1. Further, FT-IR, ^1^H NMR, and ^13^C NMR analysis suggested that the IPS-2 was mainly composed of the β-(1→4)-D-Galactose and β-(1→4)-D-Glucose as the main chain, with the β-(1→6)-D-mannose as branched chains. Compared to IPS-1, the IPS-2 showed higher antioxidant activities with an IC_50_ value of 108 ± 2.5 μg/mL, 272 ± 4.0 μg/mL, and 760 ± 5.0 μg/mL for ABTS^+^ scavenging, DPPH radical scavenging, and ferric reducing power, respectively. In addition, the IPS-2 inhibited the viability of prostate cancer (PC-3) cells (IC_50_; 435 ± 3.0 μg/mL) via apoptosis associated with mitochondrial membrane potential collapse and altered morphological features, which was revealed by cellular staining and flow cytometric analysis. Moreover, no apparent cytotoxic effects were seen in IPS-2-treated (1000 μg/mL) non-cancerous cells (HEK-293 and NIH3T3). Overall, the findings of this study suggest that *P. radiatolobatum* could be a potent source of polysaccharides with promising antioxidant and anticancer activity.

## 1. Introduction

In 2020, prostate cancer (PC) accounted for more than 1.4 million new cancer cases and almost 0.4 million deaths worldwide, making it the second most frequently diagnosed cancer and the fifth leading cause of cancer-related death among men [1]. The onset and progression of PC are positively correlated with increased oxidative stress, which is caused due to the host’s impaired antioxidant and pro-oxidant systems [2]. Generally, localized PC is treated with radical prostatectomy and/or radiation, while androgen ablation is the main treatment for advanced-stage and/or recurrent PC. In addition, for metastatic PC, abiraterone, cabazitaxel, docetaxel, enzalutamide, and Sipuleucel-T (Sip-T) are the United States Food and Drug Administration (FDA)-approved treatments; however, these treatments provide an overall survival (median) range of 13–32 months with a 5-year survival rate of only 15% [3]. The potential risk factors considered for PC and other oxidative stress-linked disorders are inflammation, androgens, diet, and lifestyle, which can be modulated by antioxidant-rich dietary components [2,4]. Although, the direct effect of antioxidant supplementation on PC remains inconclusive [5], natural antioxidants are used to minimize the side effects and improve prognosis during cancer therapy [6]. For instance, several bioactive and biocompatible small molecules and macromolecules derived from natural sources are reported to exhibit antioxidant and anticancer properties [6,7]. In this regard, polysaccharides have been proven to be potent anticancer agents due to their immunomodulatory and antioxidant behavior and thus are of great importance [7,8]. On the other hand, the global polysaccharides market size was valued at USD 14.04 billion, growing at a compound annual growth rate (CAGR) of 4.8%, and is projected to reach USD 21.41 billion by 2030 [9].

Polysaccharides are the most abundant naturally occurring and structurally diverse macromolecules, consisting of long chains of monosaccharide units connected through glycosidic linkages [7,8,10,11]. The structural features of the polysaccharides, mainly the molecular weights, are associated with their antioxidant and anticancer behavior. Conceivably, the polysaccharides with lower molecular weights contain reductive hydroxyl group terminals (per unit mass), enabling free radical acceptance and elimination [12]. On the other hand, polysaccharides with higher molecular weights are thought to exhibit stronger connections with receptors/proteins on cancer cells that perhaps exert enhanced anticancer effects [13]. Yet, several studies have reported that the polysaccharides derived from natural sources could behave as antioxidant and anticancer agents [14,15,16]. The mechanisms of the anticancer actions of polysaccharides include apoptosis, cell cycle arrest, immunomodulation, anti-angiogenesis, motility inhibition, and anti-mutagenesis [8]. However, the exact mechanism of antioxidant behaviors of polysaccharides and the precise relationship between the chemical properties of polysaccharides and their cytotoxicity mechanisms are still inadequately understood.

The sources of natural polysaccharides could be animals, plants, algae, fungi, lichens, and bacteria. However, numerous studies have reported fungal polysaccharides for their promising biocompatibility, biodegradability, cost-effectivity, enhanced solubility and stability, non-immunogenicity, and therapeutic efficacy [10,14,17,18,19,20]. Filamentous fungi could produce two kinds of polysaccharides, such as extracellular (EPS; release into the culture medium) and intra-cellular (IPS; main constituent of the cell wall (mycelia, fruiting bodies, and spores)) [21]. Among them, *Penicillium* is an important Genus comprising more than 400 identified species. Further, biomolecules of *Penicillium* species, such as alcohol, penicillin, peptides, and organic acids, are promising in agriculture, food, and medicinal industrial applications [22,23]. Some earlier works reported that IPS and EPS were isolated from *Penicillium* sp., and determined their promising biological activities [11,24,25]. However, endophytic *Penicillium* is purported to hold inherent traits of the concerned host plants and produce antagonist compounds, protecting plants against biotic stress [26]. Additionally, an endophytic *Fusarium solani* is reported to produce polysaccharides similar to those of the host plant (*Dendrobium officinale*) [27]. It is also reported that endophytic *Penicillium* spp. produce anticancer biomolecules [28]. So, endophytic *Penicillium* sp.-derived polysaccharides could be promising in anticancer applications.

Although filamentous fungi have extensive usage in the agriculture, food fermentation, textile, and pharmaceutical industries, a considerable amount of fungal mycelia are produced as by-products of industrial fermentation processes and are discarded or burned [11,18]. Thus, utilizing fungal mycelia for polysaccharide isolation could be valuable in sustainable applications. On the other hand, the bioactivities of polysaccharides are greatly influenced by their structural (monosaccharide composition, types of glycosidic linkages, branching point, and confirmation of carbon, hydrogen, and oxygen atoms), cultivar/source (origin, batch materials, extraction, and drying methods), and physicochemical (charge density, molecular weight, solubility, and polarity) properties [12]. In this way, researchers are constantly attracted to polysaccharides research while searching for novel and potent therapeutic agents. Therefore, this study intended to extract, fractionate, and characterize the IPS from the mycelia of *Penicillium radiatolobatum* and examine their antioxidant and cytotoxic activity.

## 2. Results and Discussion

### 2.1. Culture Medium Composition and IPS Production

The culture conditions and variety of nutritional contents play crucial roles in determining the synthesis, yield, composition, structure, and bioactivity of microbial polysaccharides. In submerged culture conditions, numerous nutrients have been linked to increased fungal polysaccharide production, including carbon and nitrogen sources and other elements [20]. The carbon source is essential for supplying the basic energy required for fungal growth and development, while the nitrogen source provides the biosynthesis of the primary building block of nucleic acids, proteins, and enzymes, supporting fungal metabolism and biomass production. It is reported that glucose and peptone are the most suitable carbon and nitrogen sources for enhancing IPS production by fungi, and the potato extract provides adequate amounts of nitrogen, mineral salts, and growth factors (B vitamins) in submerged fermentation [29,30], although no single cultivation medium can ensure high IPS productivity for all microbes. In numerous studies, the IPS from *Penicillium* species are produced in media enriched with potato extract, glucose, and peptone [18,24,25]. Therefore, in this study, potato-extract-, glucose-, and peptone-containing media have been used for IPS production from endophytic PR.

### 2.2. Extraction, Yield, Fractionation, and Chemical Composition of IPS

Hot water extraction is widely applied for IPS extraction from fungi because of simple operation requirements and causing minimal structural damage to polysaccharides [17]. Crude IPS was extracted from the dry mycelium of endophytic PR by hot water and precipitated through EtOH with a total dry weight of 6.37 g (Table 1). Generally, crude polysaccharides are accompanied by proteins, pigments, inorganic salts, and non-polar substances. Thus, the extracted crude IPS was deproteinized and decolorized by the Sevage technique, and the dialysis process could remove other impurities [10,31]. Furthermore, the crude IPS was fractioned by the DEAE Sepharose Fast Flow column, and the carbohydrate content in the fractions was assessed by phenol-sulfuric acid assay. A total of two fractions of IPS (IPS-1 and IPS-2) were collected (Figure 1a), dialyzed, and freeze-dried. However, the elution profile of IPS indicated a single and symmetrical peak for IPS-1 and IPS-2 (Appendix A) with a total yield of 28.42% and 19.38%, respectively (Table 1). In chemical composition assays, the TPC (mg of GAE/g) was detected at a trace level of 0.25 ± 0.05 for IPS-1 and 0.11 ± 0.02 for IPS-2. But, the TFC was not detected in the tested samples (IPS-1 and IPS-2) (Table 1). Similarly, a study reported that the TFC was not found in the IPSs isolated from the endophytic *Trichoderma harzianum* [16]. The presence of proteins was not detected in either of the samples (IPS-1 and IPS-2). Moreover, the UV-visible absorbance spectra of IPS-1 and IPS-2 indicated no absorption peaks at wavelengths of 260 and 280 nm and suggested the absence of proteins and nucleic acids in the tested samples (Table 1 and Figure 1b). The UV-Vis spectrum analyses (in the 260–280 nm range) of an earlier study also report the absence of nucleic acids, proteins, and pigments in the IPS extracted from *P. crysogenum* [11].

### 2.3. Monosaccharide Composition Assessment

It is known that polysaccharides consist of a chain of monosaccharides linked by glycosidic bonds that can be hydrolyzed by acids to determine the monosaccharide units. Utilizing an HPLC-coupled UV detection system and a mobile phase consisting of acetonitrile and phosphate buffer offers a promising approach for measuring the sugars [32]. The standard monosaccharides were analyzed through the HPLC-UV system and determined at the retention time (min) as 7.756 (mannose), 9.317 (xylose), 15.214 (glucose), 16.840 (galactose), 17.052 (ribose), and 17.784 (arabinose) (Appendix A). Compositional analysis of IPS-1 and IPS-2 was studied by matching the retention time of standard monosaccharides mix, and the resulting HPLC chromatogram is presented in Figure 2a–c. The HPLC results indicated that the retention peaks of IPS-1 were matched with the standard mix as mannose, xylose, glucose, and galactose in a concentration of 2.99%, 3.61%, 55.28%, and 38.10%, respectively (Figure 2a,b and Table 1). Similarly, the IPS-2 consisted of mannose (24.84%), glucose (4.44%), and galactose (70.71%), while xylose was not detected (Figure 2a,c and Table 1). Moreover, ribose and arabinose were not detected in the tested samples (IPS-1 and IPS-2). Although fungal polysaccharides predominantly comprised monosaccharides, including arabinose, fucose, rhamnose, ribose, xylose, mannose, glucose, and galactose [17]. The glucose content was found to be highest (55.28%) in IPS-1 compared to IPS-2 (4.44%). Meanwhile, the IPS-2 indicated the highest galactose (70.71%) and mannose (24.84%) content compared to IPS-1 (38.10% and 2.99%, respectively) (Table 1). An IPS fraction (PCPS) form *P. crysogenum* is also reported for higher mannose (59.9%) and galactose (34.3%) content with a trace amount of glucose (3.4%) and rhamnolipid (2.4%) in a previous study [11]. However, it is worth noting that the monosaccharide compositions in the fungal polysaccharides could be affected by the source of origin and genetic makeup of fungi. A previous study has reported the presence of mannose (45.5%), galactose (39%), glucose (10%), and glucuronic acid (5.5%) monosaccharides in the IPS extracted from *Trichoderma kanganensis* [14]. Similarly, the IPSs extracted from the endophytic *Trichoderma harzianum* consisted of mannose, glucose, and galactose [16].

### 2.4. FT-IR Analysis

The functional groups of the IPS-1 and IPS-2 were studied by FT-IR analysis at the wavenumber range of 4000–500 cm^−1^, as presented in Figure 3. The spectrum of IPS-1 indicated strong absorption peaks at 3293.55 cm^−1^, 2928.96 cm^−1^, 1647.42 cm^−1^, 1561.48 cm^−1^, 1407.77 cm^−1^, 1144.58 cm^−1^, 1105.53 cm^−1^, 997.02 cm^−1^, and 612.89 cm^−1^. Meanwhile, the IPS-2 exhibited strong absorption peaks at 3338.41 cm^−1^, 2926.34 cm^−1^, 1653.98 cm^−1^, 1133.91 cm^−1^, 1106.75 cm^−1^, 996.53 cm^−1^, 636.97 cm^−1^, and 614.78 cm^−1^. Taken together, the characteristic peaks at 3293.55 cm^−1^ and 3338.41 cm^−1^ were associated with the O-H stretching vibrations of the hydroxyl group, and the peaks at 2926.34 cm^−1^ and 2928.96 cm^−1^ were ascribed to C-H stretching vibrations in the methylene group [14,16,20]. In addition, the characteristic peaks at 1647.42 cm^−1^ and 1653.98 cm^−1^ were accredited to the C=O stretching vibrations [33]. However, the characteristic peaks in IPS-1 at 1561.48 cm^−1^ and 1407.77 cm^−1^ were due to C-O vibration and C-H bending vibration, respectively [24,34], but IPS-2 did not exhibit those same peaks (Figure 3). The characteristic peaks in the range of 1100–1150 cm^−1^ were attributed to the C-O stretching vibrations typically associated with primary or secondary alcohols in polysaccharides [16]. The strong peaks at 996.53 cm^−1^ and 997.02 cm^−1^ were ascribed to the C-O-C stretching vibrations [24,33]. Additionally, the peaks around 600 cm^−1^ (IPS-1 and IPS-2) were associated with the bending vibrations of a polysaccharide ring [35]. Overall, the FT-IR analysis confirmed the polysaccharide nature of IPS-1 and IPS-2 based on the presence of characteristic peaks (3293.55 cm^−1^, 3338.41 cm^−1^, 2926.34 cm^−1^, and 2928.96 cm^−1^) associated with sugars (Figure 3) [11].

### 2.5. NMR Analysis

The structural information of IPS-1 and IPS-2 was studied using NMR spectroscopy, as shown in Figure 4. The glycosidic linkages in polysaccharides could be of two different types: α-type or β-type. The hemiacetal (ketone) hydroxyl group arrangement mostly determines the configurations of these bonds. From HPLC results (Figure 2), it was found that the monosaccharide units in IPS-1 were glucose (Glc), galactose (Gal), mannose (Man), and xylose (Xyl), and the peak area values suggested the ratio is like Glc:Gal:Xyl:Man; 13:10:1:0.8. While the monosaccharide units in IPS-2 were Gal, Man, and Glc, and the peak area values suggested the ratio is like Gal:Man:Glc; 12:4:1. In the ^1^H-NMR spectrum, the chemical shifts of the IPS-2 signals were mainly distributed in δ 3.4–4.3 and δ 5.1–5.3 (Figure 4a). However, the chemical shifts of the IPS-2 signals were mainly distributed in δ 3.6–4.1 and δ 5.0–5.3 (anomeric proton region) (Figure 4b). In the ^13^C-NMR spectrum, the IPS-1 signals (ppm) were found at 99.82 (C1), 81.36 (C3), 74.13 (C5), 71.76 (C2), 70.38 (C4), 60.58 (C6) (Figure 4c). Similarly, the IPS-2 signals (ppm) were distributed at 102 (C1), 82.61 (C3), 76.4 (C5), 72.79 (C2), 70.34 (C4), 61.09 (C6) (Figure 4d). For the β configuration, the δ was found to be equal to or less than 5.0. Hence, it could predict the stable β configuration of monosaccharide units. The most possible structure of the IPS-2 is depicted in Figure 5. The IPS-2 was mainly composed of the β-(1→4)-D-galactose and β-(1→4)-D-glucose as the main chain, with the β-(1→6)-D-mannose randomly as the branched chain. However, it is generally considered that polysaccharides with β-configuration have higher activity in comparison to their corresponding α-configurations [36].

### 2.6. Antioxidant Activities

Antioxidants play a crucial role in maintaining the normal metabolism of mammals by safeguarding against oxidative stress and related health problems. Specifically, antioxidants serve as the substrate to hinder incomplete oxygen reduction and subsequent production of reactive oxygen species (ROS) in mitochondria, thereby protecting cells from oxidative damage [37]. Numerous studies suggest that fungal polysaccharides could be pivotal in delivering antioxidant effects due to their natural composition and low toxicity [17]. However, relying solely on a single antioxidant experiment is considered inadequate to predict the comprehensive antioxidant efficacy of natural compounds. Therefore, the antioxidant effects of the IPS-1 and IPS-2 were assessed through radicals (ABTS and DPPH) scavenging and ferric-reducing power assays as presented in Figure 6 and Appendix A.

#### 2.6.1. ABTS Radicals Scavenging Activity

The reduction of the blue-green ABTS^+^ chromophore (generated via potassium persulphate and ABTS reaction) into the colorless product, accompanied by absorption detection at 734 nm, is considered an indicator of the ABTS^+^ scavenging activity of an antioxidant molecule [38]. The findings of this study indicated the concentration-dependent ABTS^+^ scavenging activity of the tested samples (IPS-1, IPS-2, and AA). The highest ABTS^+^ scavenging activity was examined for AA (86.16%), IPS-1 (65.63%), and IPS-2 (70.70%), at 1000 μg/mL concentration (Figure 6a). Moreover, the IC_50_ of IPS-1, IPS-2, and AA was defined as 223 ± 2.0 μg/mL, 108 ± 2.5 μg/mL, and 54.5 ± 1.25 μg/mL, respectively (Appendix A). Similarly, the IPS isolated form *Craterellus cornucopioides* is reported to scavenge the ABTS radicals in a concentration-dependent manner with an IC_50_ value of about 150 μg/mL [19]. However, the IPS fractions of *Agaricus bitorquis* (Quél.) Sacc. Chaidam (ABSC)- ZJU-CDMA-12 is reported to scavenge the ABTS radicals but requires a high concentration dose [15]. Moreover, in a previous study, a polysaccharide derived from endophytic *Fusarium solani* contains a similar monosaccharide composition (Glc, Gal, and Man) to IPS-2, yet compared to IPS-2, their IC_50_ value (>4000 μg/mL) for ABTS radical scavenging is significantly higher [27].

#### 2.6.2. DPPH Radicals Scavenging Activity

DPPH is a stable free radical, appears purple in alcohol, and displays a prominent absorption peak at 517 nm. Antioxidant molecules can interact with the unpaired electron of DPPH, reducing the purple color into a faded or yellowish appearance and indicating the DPPH radical scavenging activity [39]. The DPPH radicals scavenging activity of IPS-1, IPS-2, and AA was found to be concentration dependent. Although, at lower concentrations (125 μg/mL), the DPPH radicals scavenging activity of IPS-1 and IPS-2 was found to be nearly equal (30%). Yet, at 1000 μg/mL, the highest DPPH radical scavenging activity of samples was observed as AA (81.81%), IPS-2 (68.42%), and IPS-1 (55.07%) (Figure 6b). The IC_50_ of IPS-1, IPS-2, and AA was defined as 643 ± 2.5 μg/mL, 272 ± 4.0 μg/mL, and 172 ± 2.5 μg/mL, respectively (Appendix A). In a previous study, the IPS fractions of *Agaricus bitorquis* (Quél.) Sacc. Chaidam (ABSC)- ZJU-CDMA-12 are reported to exhibit varied IC_50_ values of DPPH radical scavenging activity [15]. However, the IPS isolated from *Sparassis latifolia* is reported to scavenge DPPH radicals (IC_50_) at 5000 μg/mL [40]. Similarly, >5000 μg/mL of IC_50_ value is reported for DPPH radicals scavenging activity of the IPS isolated from *Fomitopsis pinicola* [34]. It is hypothesized that the composition of monosaccharides and their divergent chemical characteristics affect polysaccharides’ free radicals scavenging activity [41].

#### 2.6.3. Ferric-Reducing Antioxidant Power

The antioxidant properties of natural compounds are largely associated with their electron-donating and free radical neutralizing ability, thus contributing to their reducing powers. Therefore, determining the reducing power of natural compounds is commonly linked to their antioxidant capacity, making the ferric reducing assay a reliable approach for determining and quantifying the antioxidant activity of antioxidants [23,42]. The ferric reducing potential of IPS-1, IPS-2, and AA was positively correlated with the concentration dosage (Figure 6c). The IC_50_ values were defined as 760 ± 5.0 μg/mL and 405 ± 3.5 μg/mL for IPS-2 and AA, respectively, while the IC_50_ of ferric reducing activity of IPS-1 was found beyond 1000 μg/mL concentration (Appendix A). Similarly, earlier studies report the ferric reducing capacities of the polysaccharides isolated from fungal sources [15,42]. However, it is worth noting that the IPS fractions from the same fungal source do not necessarily reduce the ferric ions at the same rate [42], and this could be the possible reason behind the varied ferric reducing antioxidant power of IPS-1 and IPS-2 (Figure 6c and Appendix A).

### 2.7. Cell Viability Analysis

Although polysaccharides derived from natural sources are safer, bioactive, and have fewer side effects, their toxicity concerns should be taken seriously. It is conceivable that fungal polysaccharides with minimal or no toxicity could promote the growth of cancer cells. On the other hand, cytotoxic polysaccharides, in addition to their inhibitory effect on cancer cells, can negatively impact normal cells [14]. Therefore, the cytotoxic effect of IPS-1 and IPS-2 was assessed in non-cancerous (HEK-293 and NIH3T3) and cancer (PC-3) cell lines, as seen in Figure 6 and Figure 7.

#### 2.7.1. Cell Viability Analysis in HEK-293 and NIH3T3 Cells

The cytotoxic impact of IPS-1 and IPS-2 was tested in HEK-293 and NIH3T3 cells using the WST assay, and the results are shown in Figure 7a,b. Compared to untreated (control) cells, sample (IPS-1 and IPS-2) treatments (31.25–250 μg/mL) did not indicate a substantial reduction in the viability of HEK-293 and NIH3T3 cells. Even at this concentration (31.25–250 μg/mL), the IPS-2-treated cells showed >100% of cell viability of HEK-293 and NIH3T3 cells. Moreover, a previous study reports that the treatment (500 μg/mL) of polysaccharides derived from endophytic *Fusarium solani* showed >90% of HEK-293 cell viability and corroborated our findings [27]. Interestingly, even at the highest tested dose (1000 μg/mL), IPS-1 and IPS-2 did not indicate the notable reduction in the cell viability of NIH3T3 cells (Figure 7b). However, at this dose (1000 μg/mL), IPS-1 and IPS-2 indicated a slight reduction in the cell viability, with 81.4 ± 3.1% and 76.6 ± 5.1%, respectively. It is worth noting that the HEK-293 cells treated with high concentrations of polysaccharides could be challenged by high osmotic pressure around and resulting in cell death of few cells due to water loss [43,44].

#### 2.7.2. AO/EB Staining Analysis

The findings of the WST assay were evidenced using membrane-permeable AO/EB stain, observed by fluorescence microscope, and compared with untreated (control) cells (Figure 7c,d). In AO/EB staining, normal cells appear in intense green fluorescence, while apoptotic and necrotic cells indicate red, orange, and/or reddish-orange fluorescence [14]. The untreated HEK-293 and NIH3T3 cells were observed in intense green with intact cell membranes and apparent morphological features. Moreover, IPS- and IPS-2-treated HEK-293 cells did not show noticeable toxicity and were also observed as similar to control cells (Figure 7c,d). Even the samples (IPS-1 and IPS-2) treated NIH3T3 cells were seen with enhanced cell viability compared to control cells (Figure 7d). However, in a recent work, endophytic *Talaromyces purpureogenus*-derived polysaccharide fractions are also reported to enhance the viability of HEK-293 cells [44]. Taken together, the findings of cell viability analysis in HEK-293 and NIH3T3 cells (Figure 7) indicated that IPS-1 and IPS-2 were free of endotoxin contamination [27].

#### 2.7.3. Cytotoxicity Analysis in PC-3 Cells

The most common anti-cancer mechanisms of the polysaccharides include direct tumor-killing (apoptosis, anti-angiogenesis, and cell cycle arrest) and/or indirect tumor-killing effects (immunomodulation). It has been reported that fungal polysaccharides can induce apoptosis in cancer cells by activating the mitochondrial apoptosis pathway via up-regulating the expression of pro-apoptotic Bax and p53 and down-regulating the expression of anti-apoptotic Bcl-2 genes [17]. Additionally, in cancer cells, polysaccharides might promote the cytochrome-C release from mitochondria into the cytosol, enhance the level of Bax protein, maintain the Bcl-2 protein level, and raise the Bax/Bcl-2 ratio; moreover, the resultant caspase-9 and caspase-3 activation also leads to the mitochondrial dependent apoptotic pathway of cancer cells [8]. The cytotoxic impact of IPS-1 and IPS-2 towards PC-3 cells was determined through WST assay, and the findings are presented in Figure 8a. Although, both the samples’ (IPS-1 and IPS-2) treatments demonstrated concentration-dependent cellular viability reduction in PC-3 cells. Yet, a significant reduction in the cell viability (%) was found in the IPS-2 (125–1000 μg/mL) treatments. The IC_50_ of PC-3 cells’ viability was defined as 435 ± 3.5 μg/mL for IPS-2, but for IPS-1, the IC_50_ value was found beyond 1000 μg/mL concentration (Figure 8a and Appendix A). It is reported that the composition and molecular mass of the monosaccharides influence the cytotoxic activity of natural polysaccharides, and probably, therefore, the difference in cytotoxicity towards PC-3 cells was found for IPS-1 and IPS-2 [33]. Moreover, it has been observed that most of the polysaccharides derived from fungal sources exhibit selective toxicity towards non-cancerous and cancerous cells [17,18]. For instance, *Trichoderma* spp.-derived IPSs reduce the viability of breast cancer (MDA-MB) cells and mouse colon cancer (CT-26) cells but do not affect the viability of mouse fibroblast (NIH3T3) cells and human hepatocytes (LO2) [14,16].

#### 2.7.4. Fluorescent Staining Analysis

The physiological impacts of IPS-1 and IPS-2 on PC-3 cells were investigated using cellular staining assays, as shown in Figure 8b–d. In AO/EB staining, compared to control cells, the IPS-1-treated cells appeared with few apoptotic cells (orange fluorescence). Meanwhile, IPS-2-treated cells were observed with more apoptotic cells, exhibiting morphological changes, nuclear shrinkage, and membrane blebbing (Figure 8b), which was accredited to the binding of AO with damaged DNA [14]. Membrane-compromised cells can be stained with red-fluorescent intercalating dye PI, and therefore, PI was used to examine the nuclear damage in PC-3 cells. As presented in Figure 8c, most of the control cells were not stained with PI, but IPS-2-treated cells were observed with greater nuclear damage and were evident for more dead cells compared to the IPS-1 treatment. Fungal polysaccharide-mediated apoptosis in cancer cells is associated with the intrinsic mitochondrial apoptotic cascade and disrupted mitochondrial membrane potential [18,45]. Also, the mitochondrial membrane potential loss is an initial event in cell apoptosis, and that can be examined using the mitochondrial stain Rh-123. Thus, treatment(IPS-1 and IPS-2) induced mitochondrial membrane potential loss of PC-3 cells was observed through Rh-123 staining. Compared to untreated (control) and IPS-1-treated cells, IPS-2-treated cells appeared with significantly decreased fluorescence emission (Figure 8d), which indicated a substantial loss of mitochondrial membrane potential and corroborated earlier findings [16,46].

#### 2.7.5. Flow Cytometry Analysis

Flow cytometry is a highly effective tool for detecting and quantifying the level of apoptosis in a population of cells. Therefore, the IPS-2-treated PC-3 cell death distribution was investigated using flow cytometric analysis applying annexin V FITC and PI staining (Figure 9). The control (untreated) cells appeared with a higher cell population (99.7%) in the bottom left quadrant, which indicates healthy cells (Figure 9a). However, the IPS-2 (250 μg/mL) treatment caused early apoptotic cell death at 23.5%, as can be seen in the bottom right quadrant (Figure 9b). Moreover, the higher dose (IC_50_) of IPS-2 indicated the apoptotic cell death at 18.2% and necrotic cell death at 27.9% (Figure 9c). Apoptosis triggered by programmed cell-death pathways controls cancer progression, while necrosis does not regulate the proliferation but kills cancer cells [47]. However, a polysaccharide fraction (EPS-1) from *Trichoderma* sp. is reported to induce MDA-MB cells death via apoptosis and necrosis [16]. Similarly, flow cytometric analysis of a previous study reported that fungi-derived IPS fraction induces apoptosis in human breast cancer (MCF-7) and cervical cancer (Hela) cells at 1000 μg/mL of the treatment dose [13]. Taken together, the findings of cytotoxicity analysis in PC-3 cells (Figure 8 and Figure 9) indicated that the IPS-1 could induce apoptosis in PC-3 cells via nuclear damage associated with mitochondrial membrane potential collapse. Moreover, the comparative analysis of some recently investigated polysaccharides derived from micro-fungi with similar monosaccharide compositions and their tested bioactivities are summarized in Table 2.

## 3. Materials and Methods

### 3.1. Chemicals and Consumables

The chemicals and consumables used in this study are listed in the Appendix A section.

### 3.2. Fungal Culture Condition and Extraction of Crude IPS

Isolated endophytic *Penicillium radiatolobatum* strain AN003 (GenBank accession ID. MW237703) was used for the production of IPS. The *P. radiatolobatum* (PR) was cultivated in a potato dextrose broth (PDB) medium. For that, the small pieces of peeled potatoes (400 gm) were boiled in distilled water (2 L) for 20 min. The boiled potato chunks were separated (filtered) using a muslin cloth, and filtrate was collected. The potato extract was supplemented with dextrose (2%) and peptone (0.25%), and before autoclaving, the final volume was adjusted to 2 L using distilled water. Then, from a freshly grown PR (on a PDA plate), ~20 plugs (5 mm) were inoculated in the PDB media and reared in an incubator at 26 ± 2 °C and 170 rpm for 2 weeks. Following incubation, the mycelia of PR (pellet) were separated from the culture media (supernatant) using centrifugation at 8000 rpm for 20 min. The intracellular polysaccharide was extracted from PR according to an earlier report [16]. In brief, the PR mycelia were suspended in distilled water (1:10 *w*/*v*) and ultrasonically treated for 15 min, and then simmered at 120 °C for 4 h. Later, the aqueous extract was filtered by cheesecloth and concentrated using a rotary evaporator under reduced pressure at 37 °C, and subsequently precipitated using a triple volume of 95% EtOH (*v*/*v*) for 14 h at 4 °C. The precipitates were centrifuged at 10,000 rpm for 10 min, and the pellet containing crude IPS was collected.

### 3.3. Purification and Separation of IPS

The impurities (pigments and proteins) were removed from crude IPS by treating with Sevage solution (n-butyl alcohol: chloroform; 1:4) twice in a separatory funnel. The aqueous layer was collected, dialyzed against water, lyophilized, and then fractioned using DEAE Sepharose Fast Flow column (1.6 × 20 cm) chromatography, as reported previously [49]. First, the distilled water was used for elution, then the different concentration of NaCl (0.1–0.5 M) was used and the flow rate (0.5 mL/min) was adjusted. The collected fractions were tested by phenol-sulfuric acid assay, pooled separately based on the eluent (water and NaCl), and dialyzed against water. Finally, the purified fractions of two polysaccharides were labeled as IPS-1 (eluted using water) and IPS-2 (eluted using 0.1 M NaCl) and stored at 4 °C for subsequent analysis.

### 3.4. Chemical Composition Analysis

The total sugar content in IPS-1 and IPS-2 was measured using a phenol-sulfuric acid assay [50]. The total phenolic content (TPC) and total flavonoid content (TFC) were determined using Folin–Ciocalteu’s method and the aluminum chloride method, respectively [51]. Total protein content was examined using the Bradford assay and bovine serum albumin was used as the standard [52]. The nucleic acid and amino acid in the IPS-1 and IPS-2 were ascertained using the UV-visible spectrophotometer (SpectraMax^®^ ABS Plus, Molecular Devices, San Jose, CA, USA) at 200–400 nm wavelength [14].

### 3.5. Monosaccharide Composition Analysis

The monosaccharides in IPS-1 and IPS-2 were determined using high-performance liquid chromatography (HPLC) coupled with a UV detector (Agilent, 1260 infinity series, Santa Clara, CA, USA) [49]. In brief, 10 mg of each sample (IPS-1 and IPS-2) was separately dissolved in 10 mL of trifluoroacetic acid (TFA; 4 M) and hydrolyzed at 110 °C for 8 h. Later, 200 μL of the hydrolyzed standard monosaccharide solutions and/or samples were added with 240 μL of NaOH (0.3 M). Afterward, the mixture was added with 240 μL of methanolic solution of 3-methyl-1-phenyl-5-pyrazolone (PMP; 0.5 M), given a quick vortex, and incubated at 70 °C for 2 h. Following incubation, the reaction mix was neutralized by adding 240 μL of HCl (0.3 M) and cooled at room temperature. Then, the mixture was added with 1 mL of chloroform and mixed vigorously. Finally, the aqueous layer was collected and filtered through a membrane (0.22 μm) filter, and the filtrate was used for HPLC analysis [44]. The mobile phase contained phosphate buffer (0.05 M) and acetonitrile (84:16; *v*/*v*), and the flow rate was 1 mL/min. The injection volume was 20 μL and the temperature of the column was 40 °C. The standard monosaccharides, such as D-(-)-arabinose, D-(-)-galactose, D-(-)-glucose, D-(-)-mannose, D-(-)-ribose, and D-(-)-xylose, were utilized to quantify the monosaccharide composition in the tested polysaccharides (IPS-1 and IPS-2).

### 3.6. FT-IR and NMR Spectroscopic Analysis

Fourier-transformed infrared spectroscopy (FT-IR; PerkinElmer Paragon 500, Waltham, MA, USA) was utilized to ascertain the functional groups’ presence in the tested samples (IPS-1 and IPS-2) at the wavelength range of 500–4000 cm^−1^. In addition, the structure and linkages of the sample were examined using ^1^H and ^13^C Fourier-transformed nuclear magnetic resonance spectroscopy (FT-NMR; Bruker, 600 MHz, Fallanden, Switzerland). The samples (20 mg) were dissolved in D2O (0.5 mL) for FT-NMR analysis.

### 3.7. Antioxidant Activity Analysis

#### 3.7.1. ABTS Radical Scavenging Assay

The ABTS radical scavenging activity of the extracted polysaccharides (IPS-1 and IPS-2) was assessed as reported earlier [16]. In brief, the ABTS^+^ stock solution was prepared by mixing the ABTS (7 mM) and potassium persulphate (2.45 mM) in a 1:0.5 ratio and incubating overnight at room temperature in the dark. Then, ABTS+ radicals (working solution) were prepared by diluting the stock solution with EtOH (50%), and the absorbance was adjusted to 0.7 ± 0.02 at 734 nm. For the ABTS^+^ radical scavenging experiment, 50 µL of serially diluted working concentration (31.25–1000 µg/mL) of IPS-1 and IPS-2 (prepared using 2 mg/mL of stock solution) was separately added with the ABTS^+^ working solution (50 µL), making total reaction volume to be 100 µL. Meanwhile, serial concentration (31.25–1000 µg/mL) of standard ascorbic acid (AA) was also analyzed to compare the results, and ABTS^+^ solution alone was taken as an experimental control. The reaction mix was incubated for 10 min in the dark at room temperature. Finally, the absorbance of the reaction mix was read at 734 nm and, the percentage (%) of ABTS radical scavenging activity was calculated [53].

#### 3.7.2. DPPH Radical Scavenging Assay

The DPPH radical scavenging activity of the extracted polysaccharides (IPS-1 and IPS-2) was assessed as reported previously [16]. In brief, the DPPH (0.1 mM) was dissolved in methanol (100 mL) to prepare the DPPH working solution. Then, 50 µL of serially diluted concentration (31.25–1000 µg/mL) of IPS-1 and IPS-2 (prepared using 2 mg/mL of stock solution) was separately added to the 50 µL of DPPH solution, making the total reaction volume to be 100 µL. Meanwhile, the AA was taken in the same concentrations to compare the results. The reaction mix was incubated for 15 min in the dark at room temperature. The DPPH solution alone was considered an experimental control. Finally, the absorbance of the reaction mix was recorded at 517 nm and DPPH free radical scavenging (%) was computed [53].

#### 3.7.3. Ferric Reducing Power Assay

The ferric ion-reducing potential of the IPS-1 and IPS-2 was examined according to the previously described protocol [23]. In brief, 500 µL of potassium ferricyanide (30 mM) solution was mixed with 200 µL of PBS (0.6 M; pH 6.6), and then, 100 µL of varying concentration (31.25–1000 µg/mL) of IPS-1 and IPS-2 (prepared using 5 mg/mL of stock solution) was separately added and incubated at 50 °C for 20 min. Afterward, 500 µL of freshly prepared trichloroacetic acid (10%) was added to the reaction mix and thoroughly mixed. Later, 1500 µL of iron chloride (0.1%) was added to the reaction mix (total reaction volume was 2800 µL) and mixed well. Finally, the absorbance was recorded at 700 nm. The standard AA was used to compare the results.

### 3.8. Cell Viability Analysis

The cytotoxic effects of IPS-1 and IPS-2 were examined in cell lines such as HEK-293 and NIH3T3 (non-cancerous cells), and in PC-3 (prostate cancer cells) using a WST assay kit. In brief, the HEK-293 and NIH3T3 cells were grown in DMEM, and PC-3 cells were grown in the RPMI medium supplemented with FBS (10%) and PS antibiotic solution (1%). Then cells (1 × 10^4^ cells/wells) were seeded (100 µL) in 96 well plates, separately, and incubated in humidified CO_2_ (5%) atmosphere at 37 °C for 24 h. Later, the wells were treated (in varying working concentrations of 31.25–1000 µg/mL) with IPS-1 and IPS-2 (prepared using 3 mg/mL of stock solution) and placed in the CO_2_ incubator for 18 h. Next, the WST reagent (15 µL) is added to each well and incubated in the plates again for 2 h at the above-mentioned conditions. Finally, the optical density was recorded at 450 nm using a microplate reader (SpectraMax^®^ ABS Plus, Molecular Devices, San Jose, CA, USA), and the cell viability (%) was computed.

#### 3.8.1. Cellular Staining Assay

The physiological impacts of IPS-1 and IPS-2 on HEK-293, NIH3T3, and PC-3 cells were studied using fluorescent staining assays. In brief, the cell suspensions (HEK-293, NIH3T3, and PC-3 cells) were transferred (4 × 10^4^ cells) in each well of 24-well plates, separately, and grown in humidified CO_2_ (5%) atmosphere at 37 °C for 24 h. Later, the wells were treated with IPS-1 and IPS-2 (250 µg/mL) and further incubated at the above-mentioned conditions for 18 h. Following incubation, the AO/EB dual-staining (for HEK-293, NIH3T3, and PC-3 cells) and Rh-123 and PI staining (for PC-3 cells) assays were performed using a culture microscope (Olympus CKX53, Olympus Corporation, Shinjuku, Tokyo, Japan), as reported in earlier works [16,22].

#### 3.8.2. Flow Cytometry Analysis

The effect of IPS-2 treatments on PC-3 cell death stages were studied using the apoptosis assay kit (Biovision Inc., Milpitas, CA, USA) (Annexin V FITC; and PI), as reported earlier [54]. In brief, the PC-3 cells were seeded in the T-25 flask (RPMI medium) and grown in humidified CO_2_ (5%) atmosphere at 37 °C. After reaching adequate confluence (75–85%), the cells were treated with IPS-2 (in 250 and 500 µg/mL of working concentration, separately) and incubated for 18 h. Following incubation, the cells were collected, washed twice with ice cold PBS (pH 7.4) through centrifugation (2000 rpm for 5 min at 4 °C), and dispersed in Annexin binding buffer (1X,100 µL). Then, 1 µL of PI (100 µg/mL) and 5 µL of FITC (as provided in the kit) were added and incubated for 15 min at room temperature. Later, the final volume was made up by adding 400 µL of Annexin binding buffer. Finally, the cell death stages were quantified by reading the cells using a flow cytometry analyzer-II (FACSymphony A3; BD, Franklin Lakes, NJ, USA).

### 3.9. Statistical Analysis

The experiments were performed in triplicate, the results were analyzed by one-way and two-way analysis of variance (ANOVA) followed by Tukey and Duncan test at *p* < 0.05 significance stage using the GraphPad Prism 8.0 and SPSS program, and the data were expressed as mean ± SD (standard deviation).

## 4. Conclusions

In this study, two intracellular polysaccharides (IPS-1 and IPS-2) were isolated from *Penicillium radiatolobatum* strain AN003, and their physicochemical and bioactive properties were compared. Both polysaccharides (IPS-1 and IPS-2) were mainly comprised of galactose, glucose, and mannose. However, the composition of monosaccharide (%) and the pattern of glycosidic linkages of IPS-1 and IPS-2 differed. Both the IPSs exhibited noteworthy antioxidant activities (radical scavenging and reducing power). Particularly, IPS-2 showed the highest ABTS^+^ and DPPH radical scavenging and ferric reducing activities. Additionally, the IPS-2 inhibited the viability of PC-3 cells via cellular damage and apoptosis associated with mitochondrial membrane potential collapse. Moreover, no apparent cytotoxic effects were seen in IPS-2-treated non-cancer (HEK-293 and NIH3T3) cells. The findings of this study suggest that the *Penicillium radiatolobatum* strain AN003 could be a potential source of polysaccharides to develop a lead compound with antioxidant and anticancer properties. However, additional research is needed to determine the exact molecular mechanism behind the differences in the bioactivities of the isolated polysaccharides.

## Figures and Tables

**Figure 1 molecules-28-05788-f001:**
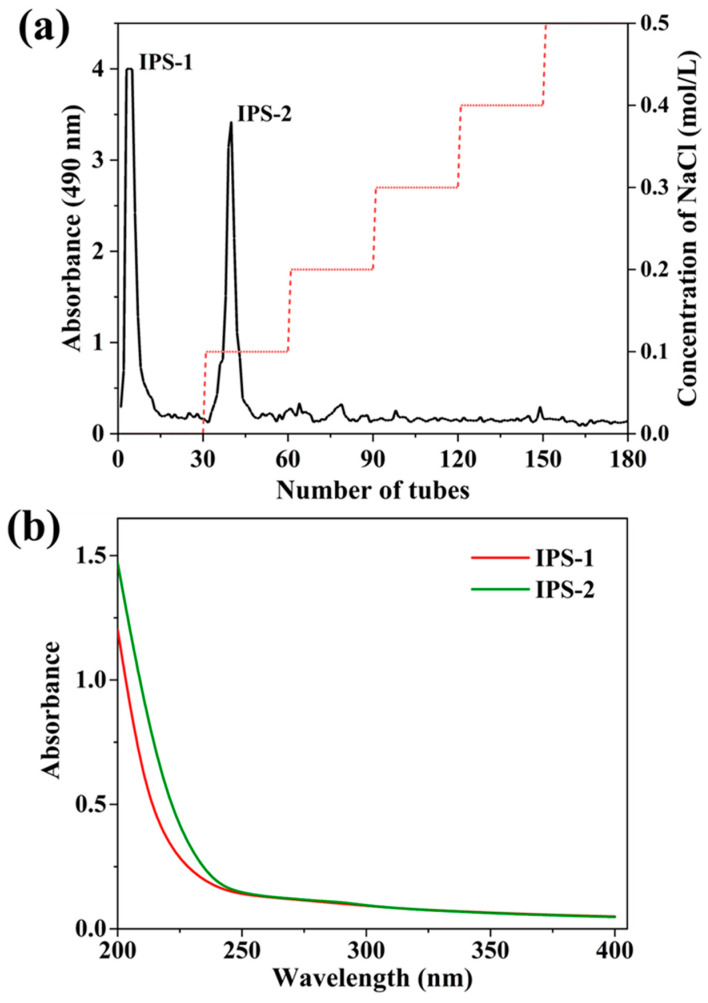
Fractionation profile of intracellular polysaccharides (IPS-1 and IPS-2) extracted from *P. radiatolobatum* by DEAE-Sepharose fast flow column chromatography eluted with different concentrations of NaCl (0, 0.1, 0.2, 0.3, 0.4, and 0.5 M), examined using phenol-sulfuric acid assay (**a**), and UV spectrum of IPS-1 and IPS-2 (**b**).

**Figure 2 molecules-28-05788-f002:**
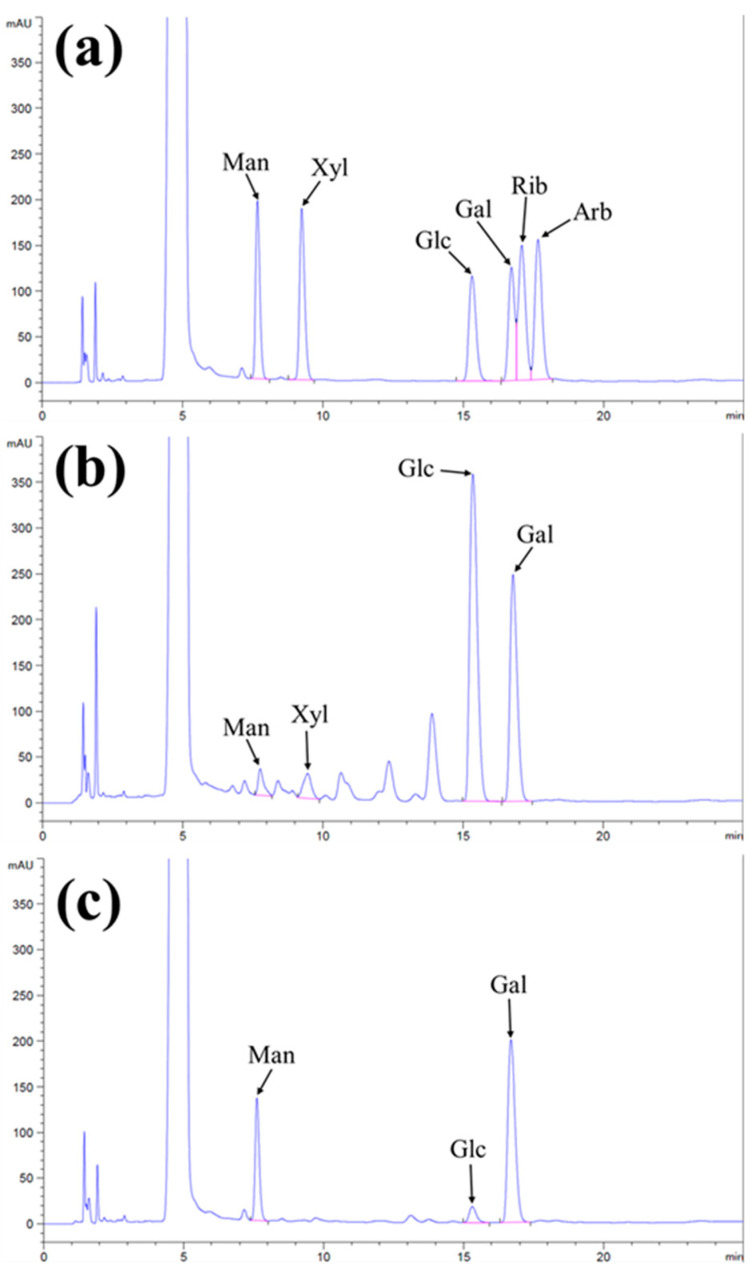
Chromatogram of standard monosaccharide mix (**a**), the composition of monosaccharides in the intracellular polysaccharides (IPS-1) (**b**), and IPS-2 (**c**) extracted from *P. radiatolobatum*, determined by using high-performance liquid chromatography (HPLC) coupled with UV detector.

**Figure 3 molecules-28-05788-f003:**
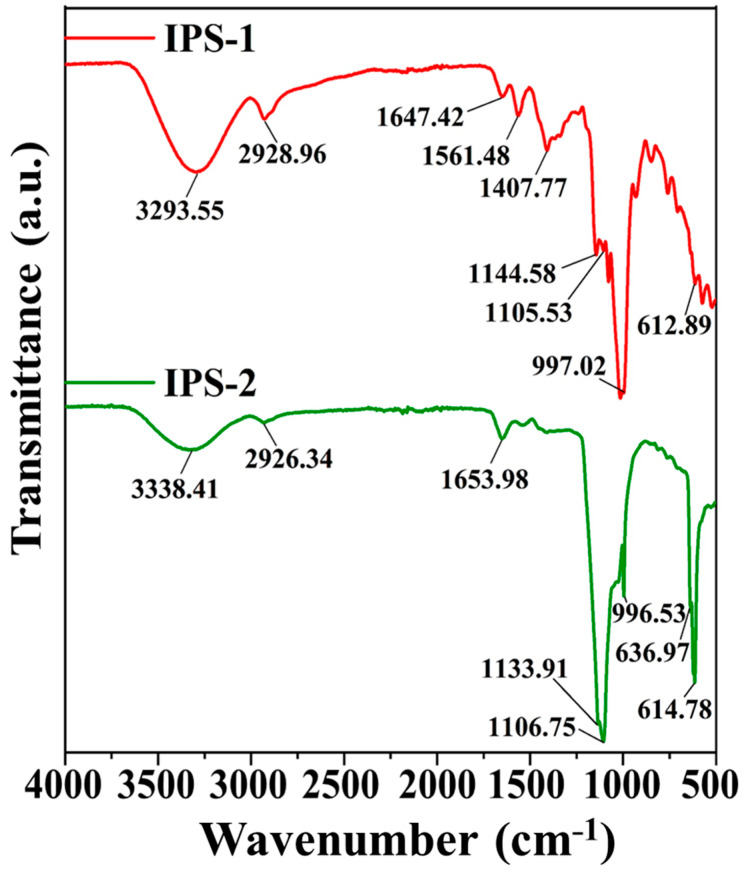
Analysis of the chemical composition and functional groups in intracellular polysaccharides (IPS-1 and IPS-2) extracted from *P. radiatolobatum* using Fourier-transformed infrared (FT-IR) spectroscopy.

**Figure 4 molecules-28-05788-f004:**
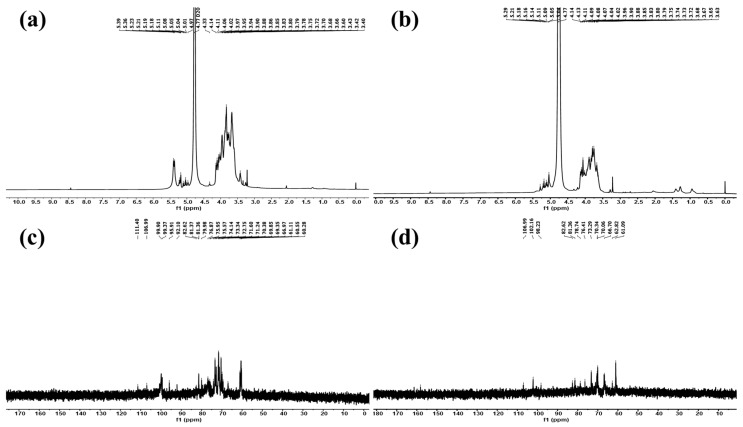
Structural characterization of the intracellular polysaccharides (IPS-1 and IPS-2) extracted from *P. radiatolobatum* using NMR spectroscopic analysis: ^1^H NMR spectra of IPS-1 (**a**) and IPS-2 (**b**) and ^13^C NMR spectra of IPS-1 (**c**) and IPS-2 (**d**).

**Figure 5 molecules-28-05788-f005:**
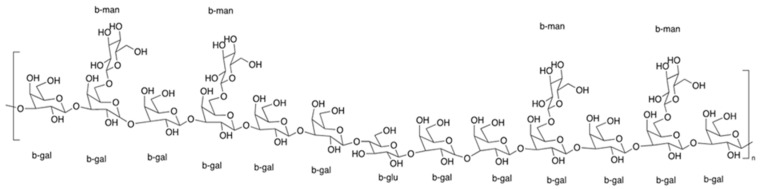
The most possible structure of IPS-2 was composed of β-D-galactose and β-D-glucose as the main chain, with the β-D-mannose randomly as the branched chain.

**Figure 6 molecules-28-05788-f006:**
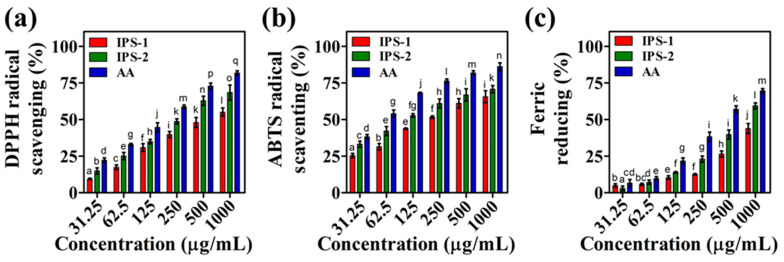
Analysis of the in-vitro antioxidant activities of intracellular polysaccharides (IPS-1 and IPS-2) extracted from *P. radiatolobatum* compared to standard ascorbic acid (AA) using ABTS scavenging (**a**), DPPH scavenging (**b**), and ferric reducing power (**c**) assay. The results are expressed as the mean, error bars indicate the SD of three independent experiments, and letters on error bars represent significant differences (*p* < 0.05).

**Figure 7 molecules-28-05788-f007:**
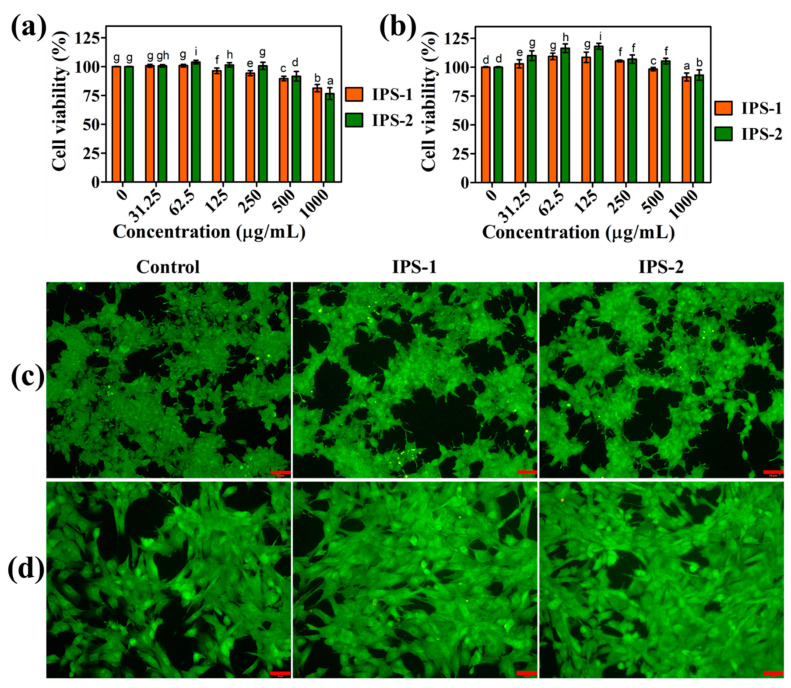
Analysis of the cytotoxic effect of intracellular polysaccharides (IPS-1 and IPS-2) extracted from *P. radiatolobatum* in human embryonic kidney (HEK-293) cells (**a**) and mouse embryonic fibroblast (NIH3T3) cells (**b**) using WST assay, and visualization of nuclear changes in HEK-293 cells (**c**) and NIH3T3 cells (**d**) using AO/EB fluorescent staining assay. The results are expressed as the mean and error bars indicate the SD of three independent experiments, and letters on error bars represent significant differences (*p* < 0.05). The images were captured at 50 µm of scale bar.

**Figure 8 molecules-28-05788-f008:**
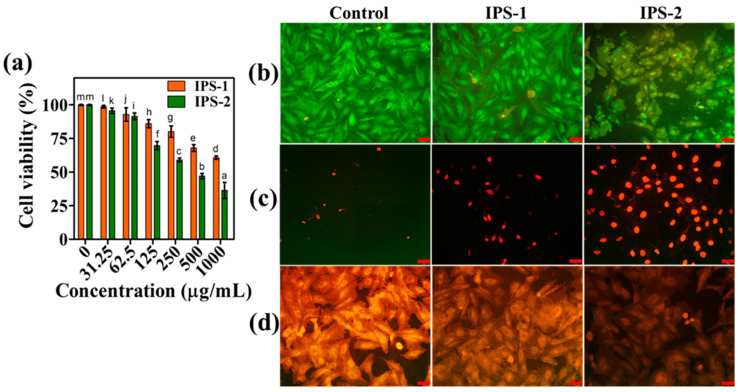
Analysis of the cytotoxic effect of intracellular polysaccharides (IPS-1 and IPS-2) extracted from *P. radiatolobatum* in human prostate cancer (PC-3) cells using WST assay (**a**) and AO/EB (**b**), PI (**c**), and Rh-123 (**d**) fluorescent staining assays. The results are expressed as the mean, and error bars indicate the SD of three independent experiments, and letters on error bars represent significant differences (*p* < 0.05). The images were captured at 50 µm of scale bar.

**Figure 9 molecules-28-05788-f009:**
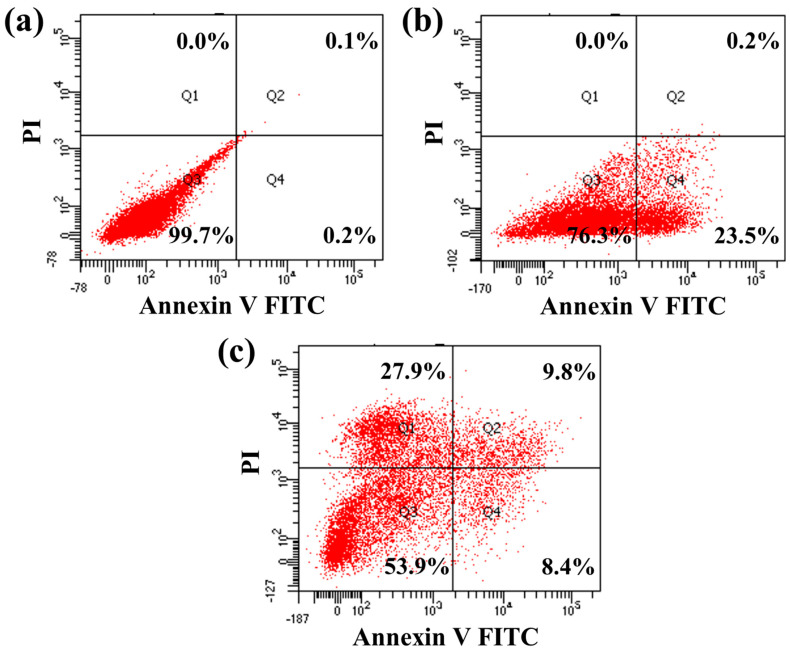
Analysis of apoptotic effect of intracellular polysaccharides (IPS-2) extracted from *P. radiatolobatum* in human prostate cancer (PC-3) cells using flow cytometry analysis, where the cells were treated with 0 µg/mL (**a**), 250 µg/mL (**b**), and 500 µg/mL (**c**) concentration of IPS-2.

**Table 1 molecules-28-05788-t001:** Analysis of the biochemical composition of intracellular polysaccharides (IPS-1 and IPS-2) extracted from *P. radiatolobatum* by spectrometric and HPLC assays. The superscript letters symbolize significant differences (*p* < 0.05).

Source	IPS-1	IPS-2
Crude weight of IPS (g)	6.37	
Yield (%)	28.42 ^b^	19.38 ^a^
Total phenol (mg of GAE/g)	0.25 ± 0.05 ^b^	0.11 ± 0.02 ^a^
Total flavonoid (mg of QE/g)	-	-
Protein (%)	-	-
Nucleic acid (%)	-	-
Monosaccharide composition (%)		
Mannose	2.99	24.84
Xylose	3.61	-
Glucose	55.28	4.44
Galactose	38.10	70.71
Ribose	-	-
Arabinose	-	-

**Table 2 molecules-28-05788-t002:** Summary of some recent investigations of the polysaccharides derived from micro-fungi and their monosaccharide composition and targeted antioxidant and cytotoxic activities.

Type of Polysaccharide (Main Fraction)	Source	Monosaccharide Composition	Tested Bioactivities (IC_50_)	Ref.
EPS (FP)	Endophytic *Fusarium solani*	Glc, Gal, and Man/MR; 2.1:3.4:3.9	ABTS radical scavenging (>4000 μg/mL); ORAC capacity (ND); cell viability [HEK-293 cells (>500 μg/mL), RAW-264.7 cells (>500 μg/mL)]	[27]
IPS (PCPS)	*Penicillium chrysogenum*	Man (59.9%), Gal (34.3%), Glc (3.4%), and rhamnolipid (2.4%)	ND	[11]
EPS (PS2-1)	*Penicillium* sp.	Glc, Gal, and Man/MR; 11.8:7.1:22.2	Radical scavenging [DPPH (>2500 μg/mL), hydroxyl (360 μg/mL), and superoxide (180 μg/mL)]; lipid peroxidation inhibition (>1300 μg/mL)	[41]
EPS (EPS) and IPS (IPS)	*Penicillium oxalicum*	EPS; Man, Gal, Glc/MR; 73.9:24.3:1.8, IPS; Man, Gal, Glc/MR; 59.1:38.9:2.0	ND	[24]
EPS (EPS)	Endophytic *Alternaria tenuissima*	Man, Gal, Glc, rhamnose, and galacturonic acid/MR; 3.25:0.95:1.0:3.02:0.45	Radical scavenging [hydroxyl (ND) and superoxide anion (ND)]; reducing power (ND)	[20]
IPS (TPS)	*Trichoderma kanganensis*	Man (45.5%), Gal (39%), Glc (10%), and glucuronic acid (5.5%)	H_2_O_2_ scavenging activity (ND); Cell viability [LO2 cells (>800 μg/mL), CT26 cells (ND)]	[14]
EPS (TP1A)	*Trichoderma* sp.	Man (18.5%), Gal (31.5%), and Glc (50.0%).	ND	[48]
IPS (IPS-2)	Endophytic *Penicillium radiatolobatum*	Man (24.84%), Glc (4.44%), and Gal (70.71%)	Radical scavenging [DPPH (272 ± 4.0 μg/mL) and ABTS (108 ± 2.5 μg/mL)]; ferric reducing power (760 ± 5.0 μg/mL); Cell viability [HEK-293 cells (>1000 μg/mL), NIH3T3 cells (>1000 μg/mL), PC-3 cells (435 ± 3.0 μg/mL)]	Present study

ND, not determined; EPS, extracellular polysaccharide; IPS, intracellular polysaccharide; IC_50_, half maximal inhibitory concentration.

## Data Availability

Data presented in this article are available on reasonable request.

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
