# Peer review of "Unveiling the Structural Characteristics and Bioactivities of the Polysaccharides Extracted from Endophytic Penicillium sp."

_molecules, 2023, doi:10.3390/molecules28155788_

Round 1

Reviewer 1 Report

The authors of the manuscript undertook the difficult task of obtaining and chemical and biological characterization of intracellular polysaccharides from endophytic Penicillium radiatolobatum. Unfortunately, the topic, however interesting and valuable, has not been properly implemented.

I regret to say that a significant part of the publication contains information loosely related to the subject of the publication. Reading the manuscript, I had the impression that the authors themselves do not fully know what they would like to convey to the readers and try to fill the word limit by force.

There is also a lack of proper discussion of the results obtained. Here, too, the authors provide data loosely related to the subject of the publication, instead of focusing on an in-depth analysis of their own results in the context of the analysis of other polysaccharides obtained from Penicillium radiatolobatum, or if such data is missing, from Penicillium sp. An alternative option would the authors could consider is a comparison of their data with results obtained from other fungal polysaccharides of similar structure. The principle is very simple, we compare like with like.

​ I appreciate that the authors of the publication are aware of the need to conduct parallel research on normal and cancer cells in order to check the safety of the tested compounds. Nevertheless, the selection of “normal cells” is incomprehensible to me for two reasons: 1) investigating the impact of compounds on prostate cancer cells, it seems natural to use as a control normal cells from the prostate, e.g. HPrEC (Human Primary Prostate Epithelial Cells) 2) furthermore, the selection of HEK-293 cell line as control was unfortunate because of the fact that these cells were transformed with adenovirus in order to the addition of Ad5 E1A and E1B genes, which immortalized these cells by interfering with cell cycle control and preventing apoptosis.

It needs to be also highlighted that the conclusions from biological studies are too far-reaching. Indeed, the authors managed to demonstrate the antioxidant effect of the tested polysaccharides, but in the case of in vitro studies, they only proved a decrease in the viability of prostate cancer cells in response to tested compounds. I cannot agree that pictures from fluorescence microscopy presented in Figure 7 provided the evidence of apoptosis induction, and does not entitle authors to make theories about the mechanism of the observed cancer cell death. I also point out the incorrect use of the terms vitality and proliferation (these are not synonymous).

​ I have serious doubts as to the method of determining the IC50 for the performed biological tests, especially after reading the authors' explanation as to the impossibility of determining the IC50 for IPS-1 based on the results of the WST assay conducted in PC-3 cells. “The IC50 of PC-3 cells’ viability was defined as 425±8.5 326 μg/mL for IPS-2, but the IC50 value could not be determined for IPS-1 up to the highest 327 tested (1000 μg/mL) concentration (Figure 7a)”.

Figure captions also require supplementation and detailing. They are too laconic, and statistical significance information is completely incomprehensible.

Extensive editing of English language required

Reviewer 2 Report

The paper is interesting and the authors performed excellent work in Unveiling the structural characteristics and bioactivities of the polysaccharides extracted from endophytic Penicillium sp. There are a few comments needed to be addressed as follows:

1-    Put an appropriate title for paragraphs from lines 91-105.

2-    Add to Figure 3 ... the chemical composition of polysaccharides.

3-    Figure No. 4 needs to improve.

4-    In Figure No. 5- Is this an analysis, explain the structures, and this will be an image.

5-    Add in introduction; what is the most used treatment for prostate cancer?

6-    Add a general mechanism of anticancer action of polysaccharides.

7-    Add a table comparing the polysaccharide extracted from endophytic Penicillium radiatolobatum in this study and other polysaccharides from other fungi and plants in prostate Cancer therapy.

8-    The discussion section is poor with the literature of the study and references; the discussion should give an interpretation of the significance of the results obtained with reference to similar works done by other authors.

Moderate editing of the English language is required.
